# Microfluidic-Assisted Human Cancer Cells Culturing Platform for Space Biology Applications

**DOI:** 10.3390/s22166183

**Published:** 2022-08-18

**Authors:** Agnieszka Krakos (Podwin), Joanna Jarosz, Patrycja Śniadek, Mateusz Psurski, Adrianna Graja, Marcin Białas, Ewa Oliszewska, Joanna Wietrzyk, Rafał Walczak, Jan Dziuban

**Affiliations:** 1Department of Microsystems, Faculty of Electronics, Photonics and Microsystems, Wroclaw University of Science and Technology, 27 Wybrzeze Wyspianskiego Street, 50-370 Wroclaw, Poland; 2Laboratory of Experimental Anticancer Therapy, Hirszfeld Institute of Immunology and Experimental Therapy, Polish Academy of Sciences, 12 R. Weigla Street, 53-114 Wroclaw, Poland; 3SatRev Company, Stabłowicka 147 Street, 54-066 Wroclaw, Poland

**Keywords:** lab-on-chip, glass micromachining, cancer cell cultivation, space biology mission, microfluidic payload

## Abstract

In the paper, the lab-on-chip platform applicable for the long-term cultivation of human cancer cells, as a solution meeting the demands of the CubeSat biological missions, is presented. For the first time, the selected cancer cell lines—UM-UC-3 and RT 112 were cultured on-chip for up to 50 days. The investigation was carried out in stationary conditions (without medium microflow) in ambient temperature and utilizing the microflow perfusion system in the incubation chamber assuring typical cultivation atmosphere (37 °C). All the experiments were performed to imitate the conditions that are provided before the biological mission starts (waiting for the rocket launch) and when the actual experiment is initialized on a CubeSat board in space microgravity. The results of the tests showed appropriate performance of the lab-on-chip platform, especially in the context of material and technological biocompatibility. Cultured cells were characterized by adequate morphology—high attachment rate and visible signs of proliferation in each of the experimental stage. These results are a good basis for further tests of the lab-on-chip platform in both terrestrial and space conditions. At the end of the manuscript, the authors provide some considerations regarding a potential 3-Unit CubeSat biological mission launched with Virgin Orbit company. The lab-on-chip platform was modelled to fit a 2-Unit autonomous laboratory payload.

## 1. Introduction

Investigation of the influence of microgravity on biological and biomedical samples has been the subject of intensive scientific works for the last decades [1,2,3]. Recently, thanks to the International Space Station (ISS) infrastructure, plenty of research facilities have been provided, e.g., Advanced Biological Research System (ABRS), BioChip SpaceLab, BioCulture System, or European Modular Cultivation System (EMCS), to ensure diverse biomedical investigation in space. Nevertheless, long waiting time, limited experiment control, and dependence on astronaut crew experience notably confine researchers to conduct specified bio-based research.

A new, lastly observed tendency is the application of a newborn CubeSats (CubeSat is a small scale satellite (*nanosatellite*) which has been used by NASA and ESA for several years in space research, mainly concerning Earth observation and more recently biomedical studies. CubeSats are standardized with cubic modules expressed in units, where 1 U ≈ 10 × 10 × 10 cm, i.e., 1.33 kg) methodology for realization of space missions, known under a common noun of bio/med nanosat technology [4,5,6,7,8,9]. Implementation of bio-medical experiments “outside” ISS, in “open” space, extends capability of research (e.g., orbit selection), cuts costs, and allows life science exploration in space. In Table 1, a summary of the aforementioned biological missions is shown [4,5,6,7,8,9,10,11,12,13,14,15,16,17,18,19,20,21,22,23].

Although the popularity of CubeSat biomedical missions is growing notably, there are still some critical issues that inhibit popularization of the CubeSat microgravity research. One of the basic limitations herein is the problem of effective biosample management due to long waiting time prior to departure. According to the rideshare launch small payloads integrators offers (e.g., Momentus or D-orbit, mostly cooperating with SpaceX as a carrier rocket supplier), integration of the payload has to be carried out circa 1–3 months before the rocket launch. This may be the reason why in the recent CubeSat missions, mammalian cells or other more sensitive biological objects have never been investigated. Additionally, often unexpected shifts (weather) and the commissioning phase of the satellite carrier, e.g., Vigoride (Momentus) and ION (D-Orbit) on the orbit (about 1–2 weeks) significantly reduce the opportunities to study living samples, e.g., human cells. There is the so-called “late access” option, which allows for a payload preparation of about 24 h before the launch but can actually be met solely with sub-orbital rocket experiments, and moreover, it does not consider further unknown mission constraints which can substantially extend the time before the fundamental part of the test starts [24,25]. Typically, CubeSats with the biological payload prior to launch are kept in the payload hangar, at ambient temperature (21 ± 3 °C). Mammalian cells in turn need an elevated temperature of incubation (37–39 °C) to ensure appropriate development of the cultures. Unfortunately, during storage, launch, and orbit transit, generally until the deployment phase, the payload cannot be operated, even on stand-by mode. It also restricts the possibility of potential culturing buffer “refreshment” with a dosing system of the payload, suggesting long-term stationary culture solely.

In the recent years, the scientific interest in the investigation of microgravity influence on human cancer cells has been especially visible. To date, few papers have been published on these biological objects and their atypical development in simulated and space-based ISS environments [26,27,28,29,30,31,32,33]. For instance, decreased proliferation, viability, and migration activity could be observed, modifying the cancer cells properties towards less aggressive way. Other reports have mentioned on the specific, multicellular spheroid growth of the thyroid and ovarian cancer cell lines in microgravity conditions. The demand for biomedical research on human cancer cells in space is strong and may shortly open the way towards novel, “orbital” oncological and antibiotic therapies, especially concerning chemotherapy and polychemotherapy.

The major proposal of the paper is to present the dedicated LOC system that would allow for cultivation of the cells in ambient conditions, without perfusion system (imitating conditions prior to rocket launch) and next, ensuring “regular” culturing experimentation, i.e., utilizing automated medium delivery system and standard incubation temperature (when the nanosatellites modules are initialized on the orbit). The key component herein is the LOC—deeply micromachined in glass substrates, which cooperates with Micro-Electro-Mechanical-System (MEMS) micropump connected to the reagent reservoir. Samples are observed by the Charge-Coupled Device (CCD) camera. Next, pictures are processed by the dedicated software and edited electronically.

Preliminary research related to cancer cell cultures on-chip has been recently carried out by the group and published in the paper [33]. Herein, two cell lines were investigated, i.e., human keratinocytes (HaCaT) and skin melanoma (A375) by culturing in LOCs under stationary conditions for 72 h. This work is an improvement over the mentioned one in the context of number of cancer cell lines for which the technology was validated, culturing duration, harsh culturing conditions applied (ambient temperature), and usage of different LOC geometry and microflow perfusion system.

## 2. Materials and Methods

### 2.1. LOC Platform

This paper focuses on the investigation of a long-term culturing of preferably human cancer cells with the use of lab-on-chip (LOC) platform. The platform is made in a form of so-called bio-payload, as shown in Figure 1.

As stated earlier in the Section 1, biological payload for CubeSat space mission must be equipped with the subsystems ensuring culture maintenance and control. Our consideration is given herein solely to the microfluidic platform fabrication, tests, and biomedical validation. The platform was prepared to provide a long-term cultivation of the chosen cancer cells in ambient temperature and to ensure automated media dosing in the typical, elevated incubation conditions. The scheme and image of the platform is shown in Figure 2.

A 15 mL falcon tube for the culturing buffer with the microfluidic adapter (model: LVF-KPT-S-2, Elveflow, Paris, France) ensuring assembly of the microfluidic fittings (model: LVF-KFI-06, Elveflow, Paris, France) is used here as a medium reservoir. Miniaturized peristaltic micropump (model: RP-Q1.5S-P45Z-DC3V, Takasago Fluidic Systems, Nagoya, Japan) is operated by the micropump controller equipped with ATtiny 48 microprocessor (Atmel, Chandler, AZ, USA) to maintain the flow of culturing media in LOC in the range of 250 µL/min. Embedded software of the micropump controller (written in AVR Assembler) manages to command up to 2 micropumps independently. The medium flow is switched herein every 24 h for 20 s. Current flow settings are presented in the LED display of the micropump controller. The power consumption of the lab-on-chip system is circa 2 W (1 W is needed for the micropump operation, and an additional 1 W is used for the micropump controller). The system is powered by 12 V power supply.

Depending on the experiment type and cancer cell line, different culturing media were used, whose specification is described in the Section 2.3. The LOC for each of the tests is placed in the 3D printed holder to provide overall protection of the structure and precise placement of the microfluidic connectors (Flangeless Fittings, Idex Health and Science, Oak Harbor, WA, USA). The observation of the cultures is carried out every 12 h with the use of a microscope optical detection system (Olympus CKX41, CCD camera SC30, Tokyo, Japan). The images of the cell colonies are acquired and analyzed utilizing the dedicated computer software (Stream Start 1.0). Depending on the experimentation cycle, the microfluidic platform is placed to the incubator (NuAire NU-5510 E, Plymouth, NA, USA) or heater (Biometra OV3, Selftec GmbH, Borken, Germany) with temperature set to 37 °C.

### 2.2. The Construction and Technology of LOC

The microfluidic chips devoted to biomedical experiments have to meet specific demands of the cells, e.g., ensure fresh culturing media dosage and oxygen diffusion to support appropriate colony growth on-chip [34,35]. For this reason, culturing LOCs are typically equipped with microchambers, microchannels and via holes. Capacity of the microchambers and microchannels is also important since it must provide sufficient space for the colony development and do not confine and impede the potential growth. Our major goal herein was to provide the most universal LOC solution, able to ensure physiological growth of as many cancer-cell lines as possible. The design of LOC is shown in Figure 3.

Based on the previous works of the authors [36,37], the geometry of the structures was optimized in the context of depth, surface to volume ratio, etc. It was assumed that the volume of the culturing microchannels is equal to 100 µL. In order to assure convenient cells inoculation, the diameter of via holes was indicated to 2 mm. The overall dimension of the culturing space was 25 mm × 3.5 mm × 0.5 mm.

As stated earlier in this article, borosilicate glass was selected as the substrate for the microfluidic chip fabrication. Apart from the certain biocompatibility, there are many additional advantages of this material, i.e., high chemical and mechanical resistance, and excellent optical properties from ultraviolet (UV) to near-infrared (IR) range that make borosilicate glass an interesting alternative to the popular polymer chips [38,39,40]. According to the report, micro-VCM test ECSS-Q-70-02 [41], glass can be also considered as one of the few materials which fulfils the low outgassing criteria for space systems.

The chip was fabricated utilizing glass micromachining techniques, i.e., wet chemical etching and fusion bonding (Figure 4).

Two borosilicate glass substrates (BOROFLOAT^®^ 33 Schott, Mainz, Germany) of standardized dimensions—50 × 25 × 1.1 mm^3^ are used herein. Utilizing CAD-based and xurography techniques, a special HF-resistive foil (Avery Dennison Graphics Solutions, Mentor, OH, USA) is CNC-cut and used to mask the slides effectively, thus selective glass micromachining can be performed. In the lower substrate, patterns of microchannels are formed, while in the upper substrate, via holes. Both of the microstructures (microchannels and via holes) are obtained in the process of deep wet isotropic chemical etching in a solution of 40% HF: 65% HNO_3_, 10:1, *v*/*v*. After the removal of the masks, the substrates are thoroughly cleaned to achieve uniform and hermetic structure thanks to the temperature fusion bonding. In this process, special washing procedures with trichloroethylene, acetone, ethanol, and Piranha are employed to prepare the substrates surfaces appropriately. Bonding of the specific temperature profile (Figure 4) is performed in a furnace, utilizing high-temperature conditions (660 °C), ensuring controllable glass reflow within the structure [37]. Utilizing this process, circa 15 structures were fabricated and tested in a reusable way for cell culturing tests. Ready to use LOCs are shown in Figure 5.

Next, the LOC platform was assembled, and its operation was verified. All the platform components exhibited appropriate performance.

### 2.3. Biological Tests Methodology

#### 2.3.1. Cell Lines and Culture Media

The human cancer cell lines used in the study are summarized in Table 2 along with the culture media used for cells culturing.

#### 2.3.2. Assessment of Various Tumor Cell Lines Ability to Grow on a LOC Platform in Normal and CO_2_-Independent Culture Media

Cells were seeded at 10^4^ and 5 × 10^4^ cells/mL density on LOC platform in medium dedicated to the cell line (Table 2). LOC was placed in the Petri dish (Greiner Bio One, Kremsmünster, Austria) and transferred to the incubator (NU-5510 E, NuAire; conditions 37 °C, 5% CO_2_). Cells were observed after 24, 72, and 120 h under light microscope (Olympus IX81, Tokyo, Japan). In the subsequent experiment cell lines that were able to grow on the LOC were slowly adapted to CO_2_-independent medium (cat. no. 18045-088; Life Technologies, Renfrew, UK), supplemented with 10% (*v*/*v*) FBS (GE Healthcare HyClone, Logan, UT, USA), 2 mM L-glutamine and antibiotics—100 µg/mL streptomycin and 100 U/mL penicillin by increasing in every passage (cells were passaged twice a week) the content of the CO_2_-independent medium in relation to the standard medium. When cells were able to proliferate in 100% CO_2_-independent medium, they were seeded at 10^4^ and 5 × 10^4^ on a LOC platform. Cells were observed after 24, 72 and 120 h under light microscope.

#### 2.3.3. Long-Term Cells Culturing in CO_2_-Free Atmosphere

Cells were seeded at 10^3^ cells/mL density on 24-well plates in CO_2_-independent medium with various concentrations of L-Gln (2 mM or 0.5 mM) and FBS (10% or 5%). 24-well plates were placed in the incubator (37 °C, 5% CO_2_) or heater (37°C *w*/*o* CO_2_). Cells were observed every 3–4 days under light microscope.

#### 2.3.4. Long-Term Cell Culturing on a LOC Platform at Room Temperature

UM-UC-3 cells were seeded at 10^4^ and 10^3^ cells on a LOC platform in an optimized CO_2_-independent medium. LOCs were placed in the Petri dish and transferred to incubator (37 °C, 5% CO_2_). After 3–7 days in the incubator, LOCs were transferred to room temperature for 14 days. Next, the temperature in the heater was gradually increased (2 °C per day), and when 37 °C was reached, LOCs were returned to the incubator for 21 days. Depending on the experiment design, the culture medium in the LOCs was manually replaced every 2–3 days or an automatic, peristaltic, pump-based system was utilized.

After 21 days in the incubator, the cells were gently harvested by trypsinization using EDTA-Trypsin solution (pH 8; HIIET, Wroclaw, Poland). Next, the cells were seeded in Petri dishes in a dedicated UM-UC-3 medium (please, refer Table 2 for details) and cultured in the incubator 37 °C, 5% CO_2_. These cells were used for subsequent antiproliferative assay and cell cycle analysis to assess if the experimental procedure influenced cells sensitivity for several standard cytostatics as well as the cell cycle profile. As a control for these assays, we used UM-UC-3 cells cultured under standard conditions.
(1)The antiproliferative assay:

A set of compounds was selected based on cytostatics diverse mechanisms of action: paclitaxel, cisplatinum, 5-fluorouracil, etoposide, gemcitabine, and doxorubicin.

Twenty-four hours prior to the addition of the tested compounds, 50 µL of cells UM-UC-3 was plated in 384-well plates (Greiner Bio One, Kremsmünster, Austria) at a density of 1 × 10^3^ cells per well. Compounds solutions were diluted in test medium—RPMI 1640 and Opti-MEM (1:1) medium, supplemented with 2 mM glutamine and 5 % fetal bovine serum FBS (Sigma-Aldrich, Chemie GmbH, Steinheim, Germany). The amount of 20 µL of tested compounds in concentration ranging (Table 3) was plated to well. The assay was read after 72 h of exposure to varying concentrations of the tested agents. The in vitro cytotoxic effect of all agents was examined using the SRB assay.

In the SRB assay, cells were fixed with cold 25% trichloroacetic acid (TCA, Sigma-Aldrich Chemie GmbH). Plates were incubated for 1 h and then washed five times with tap water. The cellular material was then stained with 0.4% sulphorhodamine B (SRB, Chemie GmbH) and dissolved in 1% acetic acid (Avantor Performance Materials Poland S.A., Poland) for 30 min. Unbound dye was removed by rinsing (4×) in 1% acetic acid. The protein-bound dye was extracted with 10 mM unbuffered Tris base (Sigma-Aldrich Chemie GmbH) and the optical density (λ = 540 nm) was determined in a computer-interfaced BioTek Synergy H4 Hybrid Microplate Reader (BioTek Instruments, Winooski, VT, USA). Results were presented as IC_50_—the concentration of tested agent which inhibits proliferation of 50% of the cancer cell population calculated utilizing GraphPadPrism 7.0 [Inhibitor] vs. response—variable slope (four parameters) nonlinear model. Each compound at the given concentrations was tested in triplicate in a single experiment. Each experiment was repeated 3–4 times.
(2)Cell cycle analysis:

10^6^ of the cells harvested by trypsinization during a standard passage was washed twice in cold PBS and fixed in 70% ethanol at −20 °C for 24 h. Then, cells were washed twice with PBS and incubated with RNAse (8 μg/mL, Fermentas, Sankt Leon-Rot, Germany) at 37 °C for 1 h, stained with propidium iodide (50 μg/mL, Sigma-Aldrich Chemie GmbH) for 30 min, and the cellular DNA content was analyzed by flow cytometry BD LSR Fortessa (Becton Dickinson, San Jose, CA, USA), using FACS Diva software (Becton Dickinson). Crude results from flow cytometry were analyzed using ModFIT 3.0 software and presented as a percentage of the cells in each cell cycle phase using GraphPad prism software.

## 3. Results and Discussion

### 3.1. Assessment of Various Tumor Cell Lines Ability to Grow on a LOC Platform in Normal and CO_2_-Independent Culture Media

At first, LOC platform fabricated as described above was evaluated for the ability to support cancer cell growth in vitro in appropriate, cell line-optimized culture medium without pump-supported medium flow. Nineteen human cancer cell lines, representing diverse growth morphology, origin and drug-sensitivity were used. Cells were seeded at 10^4^ or 5 × 10^4^ cells/chamber density, and their morphology was evaluated after 24, 72, and 120 h under light microscope for the signs of proper attachment, colonies formation, and proliferation but also for the sings of cell death (detachment, cells rounding, and debris accumulation in culture medium). Cell growth was assessed by an experienced scientist in comparison to cells cultured in standard conditions on petri dishes or in culture flasks. These observations are summarized in Table 4.

After 24 h, most of the cell lines started to attach to the growth surface with visible signs of proliferation and colonies formation in some cases (especially when a higher cells number was seeded). The rate of both processes was significantly higher in the vicinity of the via holes, plausibly because of the facilitated gas-exchange in that area. Ovary adenocarcinoma SKOV-3 cell line showed the best-looking growth image (please, see Figure 6). After some time, most of the cells were already flattened and sprawled, and the first colonies started to form. It reflects the high attachment rate observed at standard condition tests for this cell line. Other cell lines, such as A549, most of urinary bladder and colon cancer cell lines showed evident signs of attachment but without colonies formation. Cell lines such as gemcitabine-adapted UM-UC-3 exhibited cell shrinkage and blebbing with large numbers of debris in culture medium without any signs of attachment.

After 72 h, a large number of the cell lines started to proliferate properly with extensive colonies formation and encroaching of the available growth area. Cell lines, such as SKOV-3, HT-29, or UM-UC-3, characterized by high proliferation rate, covered over 40–50% of the available growth area, especially when applied at higher cell density. Chambers with cell lines (e.g., UM-UC-3 drug-resistant sublines or 5637) exhibiting negligible attachment rate after 24 h were filled with a large number of cell debris with no signs of proper cell growth. This was also true after 120 h. Most of the other cells showed at least acceptable morphology (cells flattened and spread on the growth area, no signs of blebbing nor toxic insertions in cytoplasm), especially at lower seeding density. At higher density, the cells covered almost the entire available area, and signs of proliferation suppression and/or multilayer growth accompanied by symptoms of cell death and detachment were visible. The concluding remarks from this stage are shown in Table 4.

Based on cell growth on cell line-optimized culture medium twelve cell lines were selected for further studies using CO_2_-independent medium. Stable cell growth in such culture medium is crucial since the use of pressurized CO_2_ cylinder in CubeSat nanosatellite is hard for the implementation. First, freshly thawed cells were adapted by gradual increase of CO_2_-independent medium content (0, 25, 50, and 100%) in every passage (cells were passaged twice a week). Finally, after 2–3 weeks, a stable growth was obtained for all cell lines except PC-3 and HCT-116, and thus, those cell lines were excluded from further studies. Cells were seeded at 10^4^ and 5 × 10^4^ cell/chamber, and their morphology was evaluated after 24, 72, and 120 h under light microscope for the signs of proper attachment and colonies formation and proliferation, but also for the sings of cell death (detachment, cells rounding, and debris accumulation in culture medium). Cell growth was assessed by an experienced scientist in comparison to cells cultured utilizing CO_2_-independent medium on petri dishes or in culture flasks. These observations are summarized in Table 5. Some representative images obtained during this stage can be found in Figure 6.

Most of the cells presented a slightly lower attachment efficiency after 24 h and slower growth proliferation pace though the experiment. Nevertheless, those differences were negligible and did not affect the generally positive results of this stage. The only negative exception was the LoVo cell line for which, especially at higher cells density, we observed a substantial decrease in attachment rate followed by an increase in cell death. At the same time, cell lines such as UM-UC-3 or RT-112 exhibited even better growth profile and cells morphology on the CO_2_-independent culture medium than on the cell line-optimized medium used in the previous stage.

### 3.2. Long-Term Cells Culturing in CO_2_-Free Atmosphere

On that basis, UM-UC-3 and RT-112 cell lines were selected for further studies focused on long-term cell culturing at room temperature and in a CO_2_-free atmosphere, and thus, in the conditions that will most likely be used prior to space experiments (Figure 7 and Figure 8). The CO_2_-independent medium composition was further optimized for fetal bovine serum and L-glutamine content since those two ingredients influence cell growth significantly. Additionally, cell growth with or without CO_2_ availability (referred to as incubator and heater, respectively) was evaluated for reference. At first, the tests were performed on 24-well plates, and both cell lines were seeded at 10^3^ cells/mL density. On day 6, all cells, regardless of the culture conditions applied, exhibited good morphology with cell line-related signs of proper attachment, colonies formation, and proliferation (extended, fibroblastic share for UM-UC-3, and polygon cell shape with mosaic shape colonies for RT-112). Images taken at day 11 showed significant influence of L-glutamine content on cells growth and morphology (especially visible for UM-UC-3 cells). Glutamine concentration 0.5 mM was generally recognized as slightly too low to support stable cells growth, and their proper morphology, especially when used along with 5% fetal bovine serum content without CO_2_ presence. In the nutrients-richest medium containing 10% FBS and 2 mM L-Gln, cells of both cell lines started to detach at day 11 (especially when cultured in incubator). Finally, culture medium containing 5% FBS and 2 mM L-Gln was selected for the further studies with LOC platform as a medium supporting proper cells morphology with limited cell proliferation pace.

Further studies with UM-UC-3 and RT-112 were undertaken utilizing the LOC platform (Figure 9, Figure 10 and Figure 11). Two distinct environments were tested: heater (with 37 °C *w/o* CO_2_) and incubator (37 °C w/CO_2_). For both cell lines, we were able to keep a steady cell growth with satisfactory morphology for over 30 days (54 days in best-case scenario). After the first 14 days of stable growth, the state of both cell lines started to deteriorate (particularly visible for RT-112 cell line cultured both in heater and incubator) with an increased level of detached cells; but after several additional days, cells image improved, and even after a long time, they still presented good morphology (especially UM-UC-3 cells).

### 3.3. Long-Term Cell Culturing on a LOC Platform at Room Temperature and in the Incubator

Based on those observations, UM-UC-3 cell line was selected for further studies utilizing LOC platform at room temperature firstly. Initially, the LOCs were placed in the incubator for 2–3 days to promote proper cells attachment (which was confirmed after 24 h). After that, LOCs were transferred to room temperature. After an additional 24 h, cells morphology started to deteriorate with a characteristic increasing amount of rounded and/or detached cells. Limited signs of proliferation were observable. However, after 14 days of the experiment, we still observed some properly attached cells, and thus, the temperature was gradually increased (~2 °C every 24 h) to 37 °C, and eventually LOCs were transferred to the incubator. After 4–5 days in the incubator, cells started to proliferate, resembling the proper morphology until the end of the experiment (after 21 days in the incubator) when a tightly packed cells monolayer was obtained (Figure 12).

Those cells were harvested by trypsinization and transferred to normal culture conditions (standard culture medium), and their drug-sensitivity and cell cycle profile was compared with UM-UC-3 cell continuously cultured in standard conditions. As can be seen in Figure 13, no significant differences were observed in cells sensitivity for six diverse drugs (the lack of differences was confirmed by the extra sum-of-squares F test for hill slope and IC_50_ with *p* < 0.05). Only marginal differences were observed in cell cycle distribution between cell lines (Figure 14; without statistical significance). Taken together, those results indicate that the experimental culturing encompassing culture at room temperature, temperature switch, and cells growth at 37 °C did not influence long-term cells abilities to growth, their drug-sensitivity, and proliferation efficiency.

Based on the previously developed procedure, an experiment with peristaltic pump-driven medium exchange in a closed loop system (‘fresh’ and ‘used’ culture media were stored in the same container) was undertaken. After proper initial cells attachment, the peristaltic pump was attached, and we were able to sustain cells growth in the incubator for about 5 days (Figure 15).

In reference to Figure 15, cells morphology differed in comparison to previous tests: cells were rounded with membranes surrounded by some number of debris; nevertheless, the cells growth was sustained with relatively high percentage of cells attachment. Although the micro-flow culture of the human cancer cells still needs to be improved, those two factors seem crucial for further studies in microgravity.

One of the modifications of the microflow system to enhance proper cells growth could be at first the application of more physiological smaller flow rate value (<250 µL/min)—not introducing excessive shear forces which can influence cells morphology badly. Different micropumps could be used for this purpose, e.g., RP-TX, Takasago Fluidic Systems, Tokyo, Japan, which operates at nL-µL range. Other changes could be related to LOC structure. For instance, microchannel depth could be bigger, ensuring greater space for colony growth. A further suggestion is to apply a matrix of small via holes coupled with a microchannel to allow for more effective gas exchange within the culture area. This solution could prevent the cells from hypoxia and related unphysiological growth.

### 3.4. The First Polish Biological Nanosatellite Mission with Microfluidic Payload Ensuring Cancer Cell Culture—Concept

Based on our preliminary experimental results, it seems that our cultivation methodology and LOC platform could be used as a microfluidic payload ensuring cancer cell development in a nanosatellite space mission. In Figure 16, a model of the 2U payload is shown, with a view to the possible arrangement of the LOCs, micropumps, falcon, cameras, etc. All the components were assembled utilizing a so-called scaffold, designed to be fabricated out of Poly-ether-ether-ketone (PEEK) material. In addition to the culturing system presented herein, a miniaturized CMOS camera equipped with a motorized lens is used, as well as a white LED lightning to appropriately view the images of the sample. In order to ensure temperature settings optimal for cancer cell development on-chip (37 °C), flexible PCB heaters are proposed to be attached within the LOCs and falcon areas. Temperature sensors are also placed within the LOCs, in etched cavities. Sealed payload housing, isolated from the external satellite structure, is fabricated out of polished aluminum, to protect the internal experiment from the mechanical damage, vibration, and harsh space atmosphere (temperature changes, radiation flux, and vacuum) during the entire mission time. The outer surface of the housing is adapted for integration into the cubic shape of the satellite platform in the CubeSat standard. Inside the payload, the sensors of temperature, humidity, pressure, and radiation are placed as well. Data exchange with satellite base modules is carried out utilizing CAN (Controller Area Network) bus and Cubesat Space Protocol (CSP). Next, the experimental data are transferred to the Earth Ground Station (GS) for further analysis. The nanosatellite overall structure covers a 3U CubeSat standard, and its mass does not exceed 4 kg. The power consumption of the CubeSat should not exceed 8 W. Our payload arrangement allows for simultaneous experimentation with three LOC structures in space.

Table 6 shows an overview of the potential biological nanosatellite mission phases and examples of environmental conditions. The most suitable approach and the shortest time of satellite integration was offered by the Virgin Orbit company. The requirements herein seem to positively correspond with our experimental assumptions and results. The lower temperatures faced potentially in the phases 0 and 1 (e.g., 4 °C) can be mitigated by the power supply provided on request by the integrator [42]. In the phase 3 (−120 °C), an operational heater of the nanosatellite base can be used to maintain proper experimental conditions. Too high temperatures of this phase (120 °C) can be in turn minimized by adequate insulation implemented by the payload housing. However, each step and energy transfer must be re-simulated and validated based on the thermal capacities of the components prior to the mission.

## 4. Conclusions

In the paper, the LOC platform ensuring long-term cultivation of human cancer cells in conditions fulfilling the requirements of the CubeSat biological missions is shown. The LOC platform was fabricated utilizing materials (e.g., glass) that are considered to satisfy low outgassing criteria in space systems subject [41]. Other miniaturized components were also applied to allow for cell culturing system miniaturization and assure appropriate cell cultivation conditions, especially with a view to the scalable perfusion system.

Nineteen cancer cell lines of different origin (e.g., A498—kidney carcinoma) were tested to verify the biocompatibility of the microfluidic platform. Most of the cells showed at least acceptable morphology, especially cell lines, such as SKOV-3, HT-29, or UM-UC-3 started to form the colonies extensively, covering circa 50% of the available growth area. Next, twelve cancer cell lines of the best morphology were chosen for the tests assuming preparation of the most optimal culturing buffer (without CO_2_). This step was necessary to be consistent with the potential culturing conditions which are faced on the orbit. Herein, similarly as in the previous experiment, acceptable results were obtained for all of the cell lines (except for LoVo), but the best growth was observed for the two lines, UM-UC-3 and RT-112.

The stationary culture of UM-UC-3 and RT-112 utilizing the LOC platform for both of the cell lines was maintained for up to 50 days (RT-112, 41 days; UM-UC-3, 52 days). Nevertheless, better culture growth rate and cells morphology were represented by UM-UC-3. For this reason, this line was chosen exclusively for the experiments conducted in the ambient temperature and next, with the perfusion system. The cells were cultured in a temperature of 22 °C for over 14 days. Although some limited signs of proliferation were noticed, after the LOCs were transferred to the incubator, the cells started to develop, resembling a proper morphology. Eventually, a tightly packed cells monolayer was obtained until the end of experiment (21 days), which can be considered as sufficient waiting time prior to the rocket departure and CubeSat mission.

In the 5-day tests with pump-driven medium exchange and a typical incubation temperature (37 °C), an acceptable culture development was observed. Longer tests, unfortunately, showed that cells morphology starts to degenerate, which can be the result of improper medium flow value (too big, too often refreshed) and/or insufficient gas exchange. This step requires further optimization works that will be addressed during the development of the final laboratory platform as a CubeSat experimental module.

In conclusion, a new insight into comprehensive preparation of the biological nanosatellite missions with sensitive, mammalian cells (human cancer cells) utilizing the LOC platform was presented. Herein, both the wait for the rocket launch and a start of the regular experiment in orbit were taken into account and treated as an individual experimental stages. In these studies, we successfully optimized most of the conditions for a Low-Earth-Orbit biological experiment utilizing UM-UC-3—human urinary bladder transitional cancer cell line cultured in vitro. To the best knowledge of the authors, in the recent CubeSat missions, mammalian cells or other more sensitive biological objects have never been investigated, and our research is the first such approach which soon may notably increase the number of biological missions with a focus put on cancer investigation and novel therapies development.

## Figures and Tables

**Figure 1 sensors-22-06183-f001:**
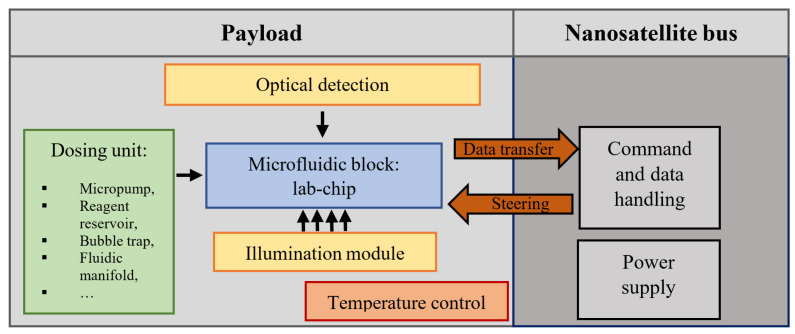
Overall schematic view of the payload for CubeSat type nanosatellite implementing biomedical experimentation.

**Figure 2 sensors-22-06183-f002:**
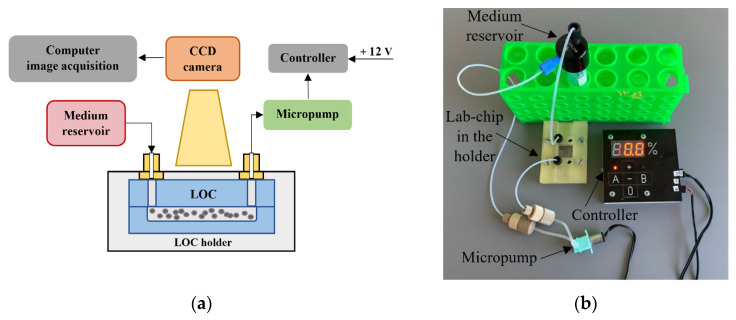
LOC platform: (**a**) detailed scheme, (**b**) platform under operation.

**Figure 3 sensors-22-06183-f003:**
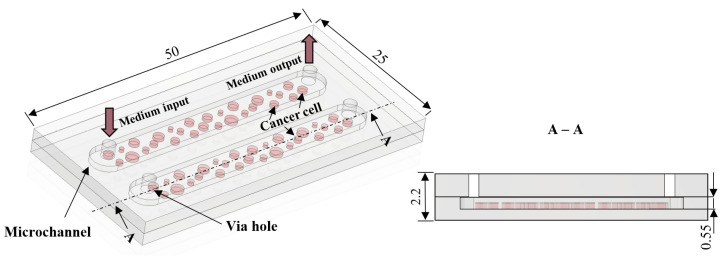
The concept of the LOC for culturing of human cancer cells: top view (on the **left**), cross-section (on the **right**). Dimensions are given in millimeters.

**Figure 4 sensors-22-06183-f004:**
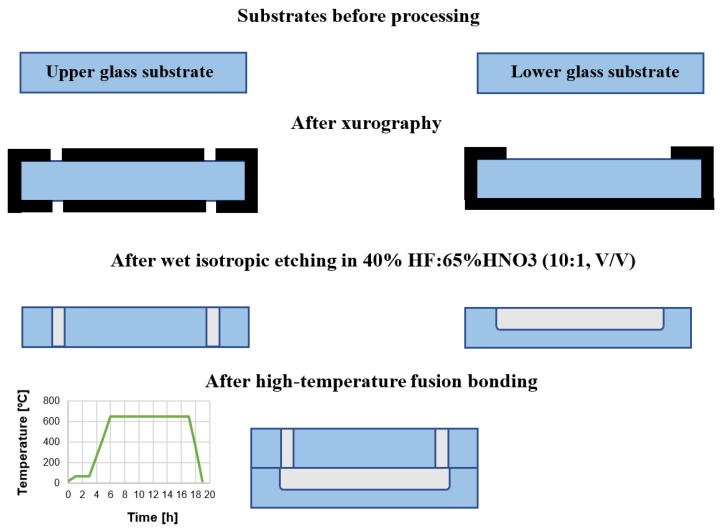
Fabrication of the LOCs for culturing of cancer cells—technology flow.

**Figure 5 sensors-22-06183-f005:**
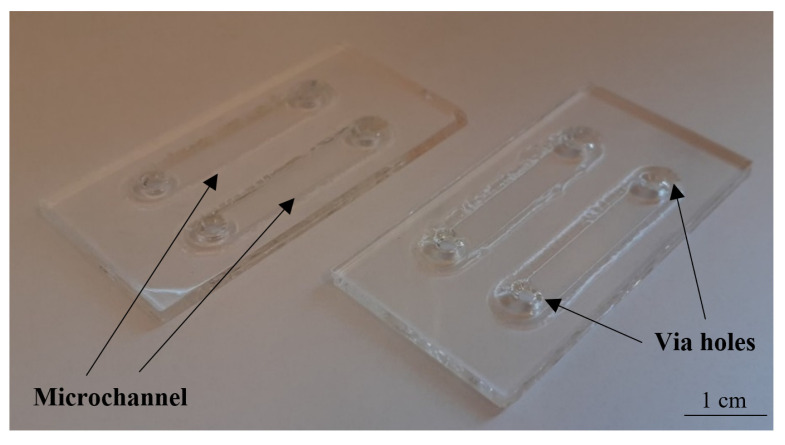
All-glass LOCs at a glance.

**Figure 6 sensors-22-06183-f006:**
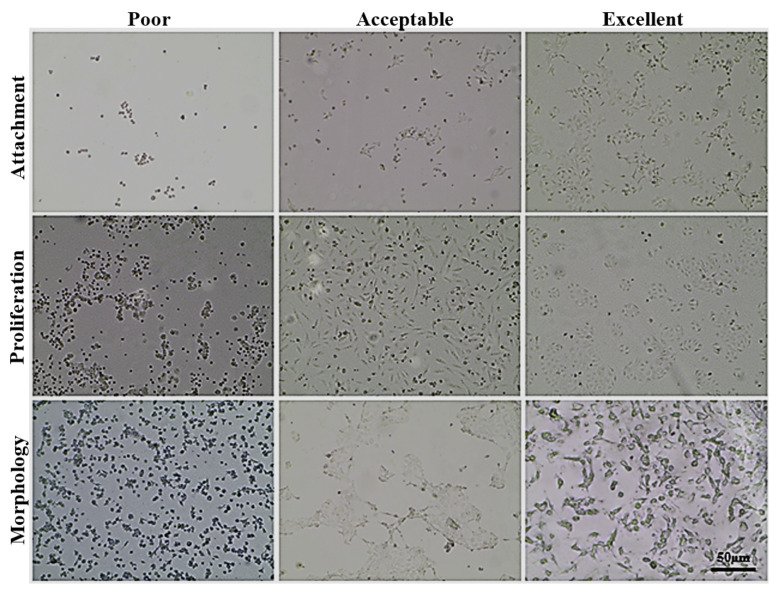
Representative images showing distinct cells attachment efficacy on LOC (mainly analyzed after 24 h), proliferation efficiency, colonies formation, cells morphology, and signs of cell death, such as cells rounding and detachment (analyzed after 72–120 h).

**Figure 7 sensors-22-06183-f007:**
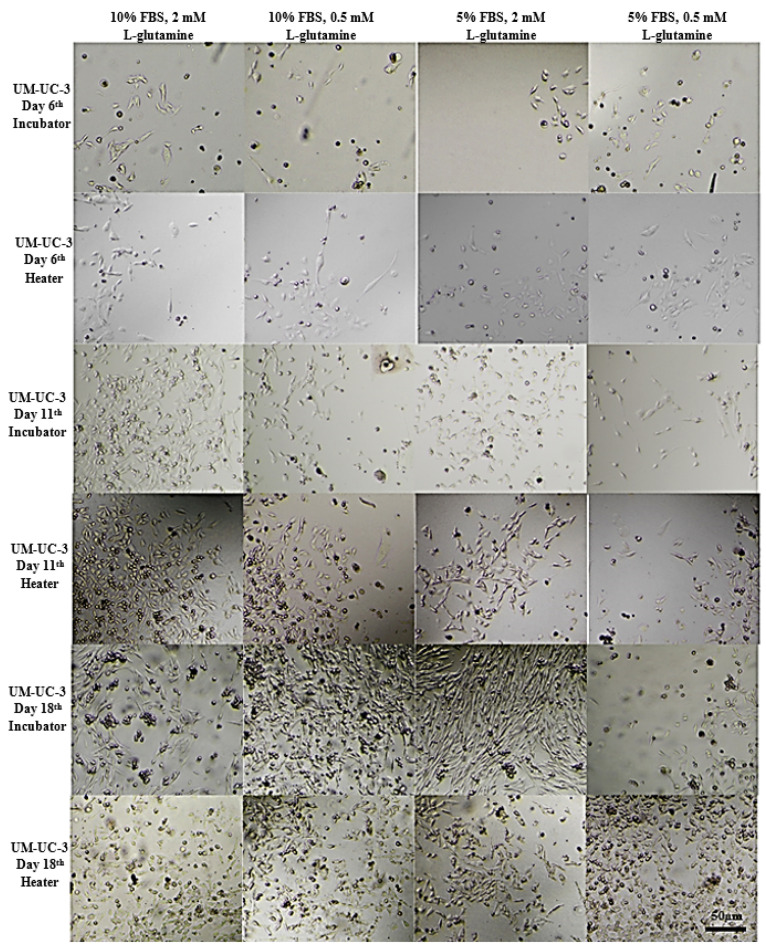
Representative images showing UM-UC-3 cells morphology at given conditions after indicated days of incubation. Incubator 37 °C, 5% CO_2_ (standard growth conditions); heater 37 °C.

**Figure 8 sensors-22-06183-f008:**
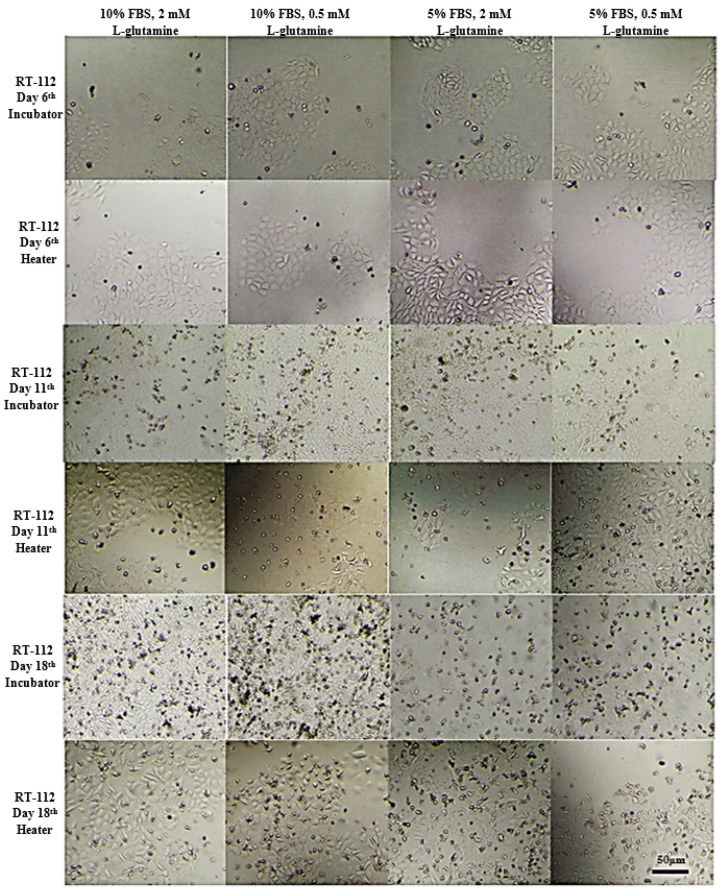
Representative images showing RT-112 cells morphology at given conditions after indicated days of incubation. Incubator 37 °C, 5% CO_2_ (standard growth conditions); heater 37 °C.

**Figure 9 sensors-22-06183-f009:**
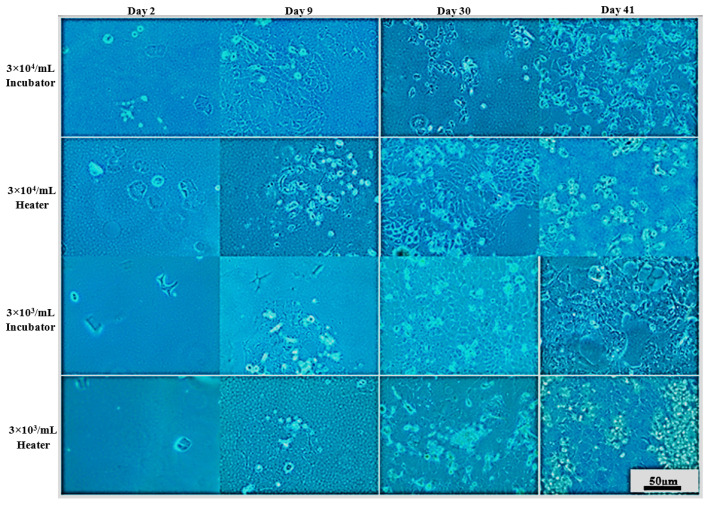
Representative images showing RT-112 cells morphology seeded on LOC platform at given conditions after indicated days of incubation. Incubator 37 °C, 5% CO_2_ (standard growth conditions); heater 37 °C.

**Figure 10 sensors-22-06183-f010:**
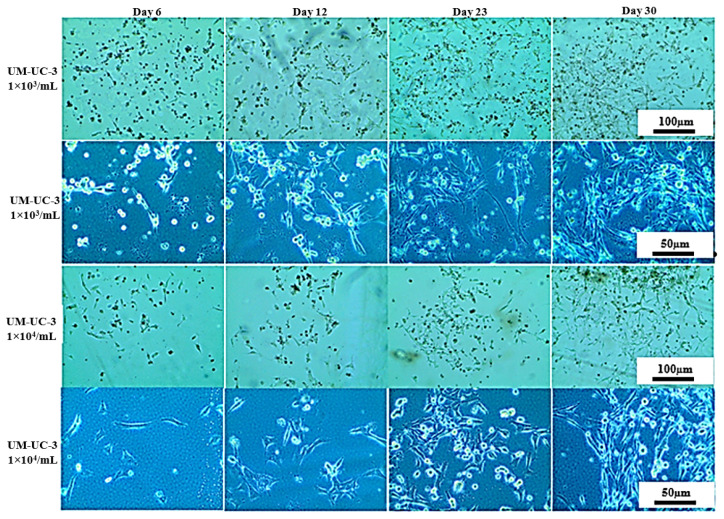
Representative images showing UM-UC-3 cells morphology seeded on LOC platform and cultured in the incubator 37 °C, 5% CO_2_ (standard growth conditions).

**Figure 11 sensors-22-06183-f011:**
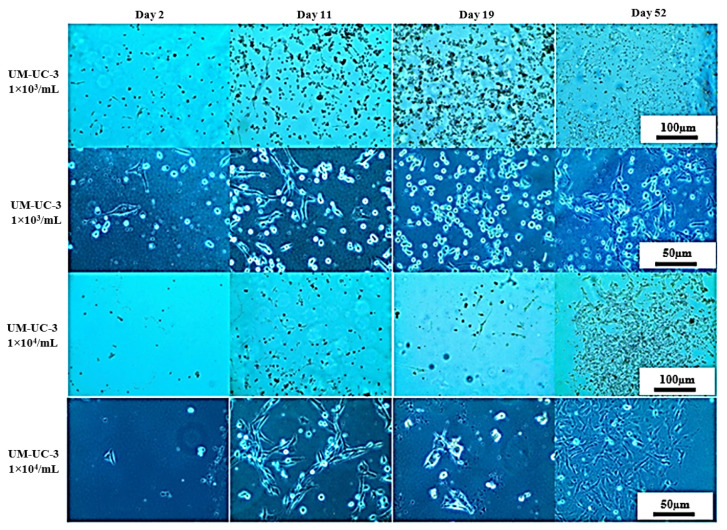
Representative images showing UM-UC-3 cells morphology seeded on LOC platform and cultured in the heater 37 °C *w*/*o* CO_2_.

**Figure 12 sensors-22-06183-f012:**
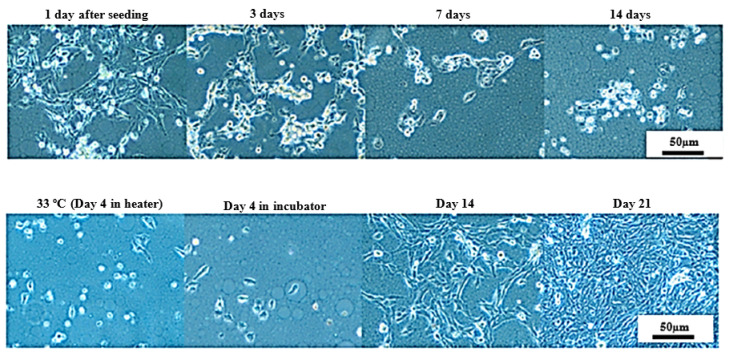
Representative images showing UM-UC-3 cells morphology cultured at room temperature for 14 days (upper panel) and after transfer to the incubator (lower panel).

**Figure 13 sensors-22-06183-f013:**
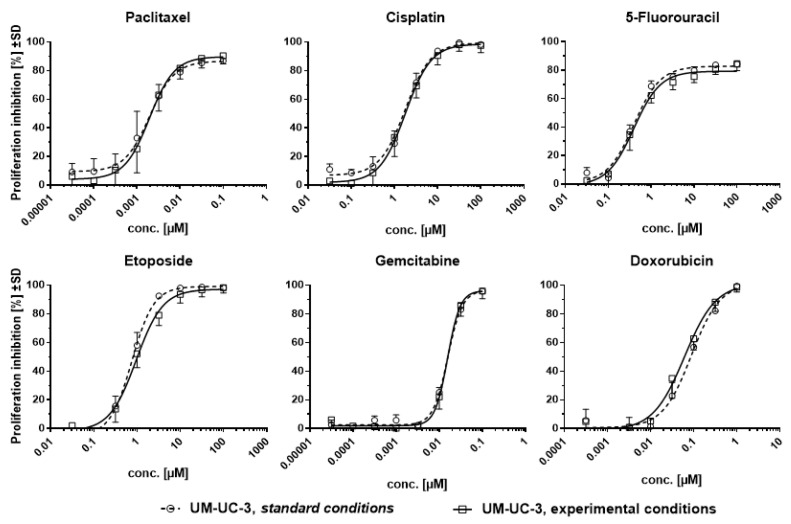
Dose–response curves obtained for UM-UC-3 cells cultured in standard or experimental conditions for 72 h with selected drugs. Curve fitting was carried out using [Inhibitor] vs. response-Variable slope (four parameters) nonlinear model in GraphPad Prism 7.0 software.

**Figure 14 sensors-22-06183-f014:**
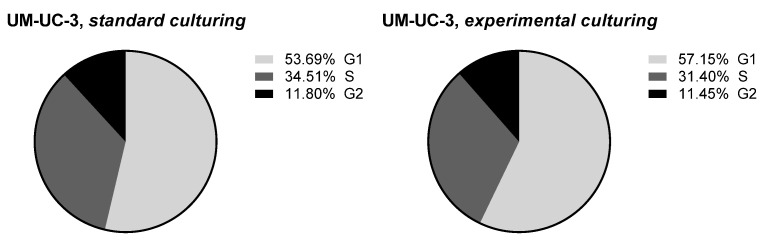
Cell cycle profile observed for UM-UC-3 cells cultured in standard or experimental conditions. Data presented as a percentage of the cells in each phase of the cell cycle. G_1_ stands for the cell growth stage, S for the DNA synthesis stage, and G2 for pre-mitosis stage.

**Figure 15 sensors-22-06183-f015:**
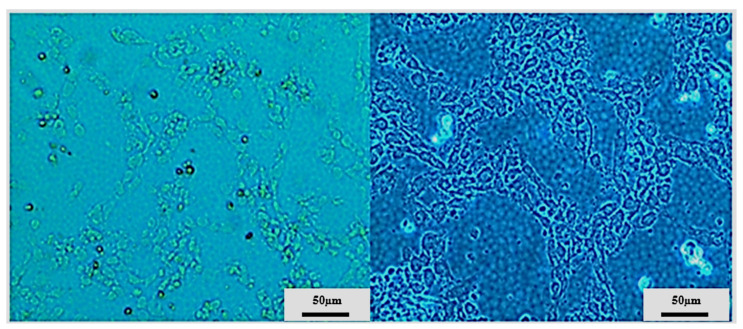
Representative images (Day 5) showing UM-UC-3 cells morphology cultured on LOC platform using peristaltic pump-driven medium exchange.

**Figure 16 sensors-22-06183-f016:**
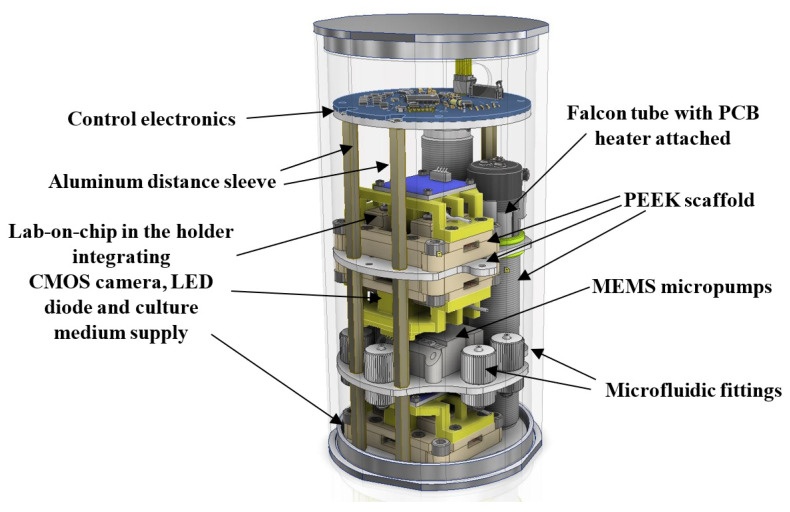
Microfluidic 2U payload for the biological nanosatellite cancer cells mission—concept.

**Table 1 sensors-22-06183-t001:** CubeSat type biological missions—overview.

	GeneSat-1	PharmaSat	O/OREOS		SporeSat	EcAMSat	Biosentinel
Nanosatellite configuration	2 U payload, 1 U bus (6.8 kg)	2 U payload, 1 U bus (5.5 kg)	2 × 1 U payloads, 1 U bus (5.5 kg)		2 U payload, 1 U bus (5.5 kg)	3 U payload, 3 U bus (14 kg)	4 U payload, 2 U bus (14 kg)
Experiment type	Gene expression of *E.coli*	Drug response of *S. cerevisiae*	Viability of *B. Subtilis* and *H. Chaoviatoris*	Chemical degradation (PAH, amino acid, porphyrin, quinone)	Gravity sensing of plant spores	Drug response of *E. coli*	DNA damage of *S. cerevisiae*
Detection methods	GFP fluorescence	Optical absorbance	Colorimetry (dye indicator)	UV–VIS spectroscopy	Conductivity of spores	Colorimetry (dye indicator)	Colorimetry (dye indicator)
Launch date	2006	2009	2010		2014	2017	2022

**Table 2 sensors-22-06183-t002:** Cell lines and culture media used in the present study.

No	Cel Line	Type of Cancer	Purchased from	Medium *
1	A549	lung carcinoma	European Collection of Authenticated Cell Cultures (ECACC, Porton Down, UK)	RPMI 1640 + OptiMEM medium (1:1) (HIIET, PAS, Wroclaw, Poland) with 5% (*v*/*v*) fetal bovine serum (FBS; GE Healthcare HyClone, Logan, UT, USA) and 2 mM L-glutamine (Sigma-Aldrich Chemie GmbH, Steinheim, Germany)
2	A498	kidney carcinoma	American Type Culture Collection (ATCC, Manassas, VA, USA)	Opti-Mem + GlutaMax (Invitrogen, Waltham, MA, USA) and RPMI1640 + GlutaMAX (Life Technologies, Renfrew, UK) (1:1) medium with 5% (*v*/*v*) FBS (GE Healthcare HyClone, Logan, UT, USA), 1 mM sodium puryvate (Sigma-Aldrich Chemie GmbH, Steinheim, Germany)
3	5637	urinary bladder TCC	Riken BRC Cell Bank	RPMI1640 + GlutaMAX with 10% (*v*/*v*) FBS (Sigma-Aldrich Chemie GmbH, Steinheim, Germany)
4	RT-112	urinary bladder TCC	RCCL (Resistant Cancer Cell Line Collection)	Dulbecco’s Modified Eagle Medium (DMEM; Life Technologies, Renfrew, UK) with 10% (*v*/*v*) FBS (GE Healthcare HyClone, Logan, UT, USA) and 2 mM L-glutamine
5	TCC-SUP	urinary bladder TCC	German Collection of Microorganisms and Cell Cultures (DSMZ, Braunschweig, Germany)	DMEM with 10% (*v*/*v*) FBS (GE Healthcare HyClone, Logan, UT, USA) and 2 mM L-glutamine
6	UM-UC-3	urinary bladder TCC	European Collection of Authenticated Cell Cultures (ECACC, Porton Down, UK)	DMEM with 10% (*v*/*v*) FBS (GE Healthcare HyClone, Logan, UT, USA) and 2 mM L-glutamine
7	UM-UC-3/CDDP	urinary bladder TCC, resistant to cisplatin	ECACC, established at Hirszfeld Institute of Immunology and Experimental Therapy of the Polish Academy of Sciences (HIIET, PAS, Wroclaw, Poland	DMEM with 10% (*v*/*v*) FBS and 2 mM L-glutamine, additionally supplemented with 2.5 µg/mL cisplatin (Accord, Warsaw, Poland)
8	UM-UC-3/GEM	urinary bladder TCC, resistant to gemcitabine	ECACC, established at Hirszfeld Institute of Immunology and Experimental Therapy of the Polish Academy of Sciences (HIIET, PAS, Wroclaw, Poland	DMEM with 10% (*v*/*v*) FBS (GE Healthcare HyClone, Logan, UT, USA) and 2 mM L-glutamine, additionally supplemented with 500 nM gemcitabine (Sigma-Aldrich Chemie GmbH, Steinheim, Germany)
9	UM-UC-3/VBL	urinary bladder TCC, resistant to vinblastine	ECACC, established at Hirszfeld Institute of Immunology and Experimental Therapy of the Polish Academy of Sciences (HIIET, PAS, Wroclaw, Poland	DMEM, supplemented with 10% (*v*/*v*) FBS and 2 mM L-glutamine, additionally supplemented with 5 nM vinblastine (Sigma-Aldrich Chemie GmbH, Steinheim, Germany)
10	HCT116	colon carcinoma	American Type Culture Collection (ATCC, Manassas, VA, USA)	McCoy’s 5 A medium (Life Technologies, Renfrew, UK) supplemented with 10% (*v*/*v*) FBS (GE Healthcare HyClone, Logan, UT, USA)
11	HT29	colon adenocarcinoma	American Type Culture Collection (ATCC, Manassas, VA, USA)	RPMI 1640 + OptiMEM medium (1:1) supplemented with 5% (*v*/*v*) FBS (GE Healthcare HyClone, Logan, UT, USA), 2 mM L-glutamine and 1 mM sodium pyruvate
12	LoVo	colon adenocarcinoma	American Type Culture Collection (ATCC, Manassas, VA, USA),	F-12K Nutrient Mixture (F-12K; Corning, Corning, USA), supplemented with 10% (*v*/*v*) FBS (GE Healthcare HyClone, Logan, UT, USA)
13	LoVo/DX	colon adenocarcinoma, resistant to doxorubicin	American Type Culture Collection (ATCC, Manassas, VA, USA)	F-12K Nutrient Mixture, supplemented with 10% (*v*/*v*) FBS additionally supplemented with doxorubicin 100 ng/mL (Accord, Warsaw, Poland)
14	A2780	ovary carcinoma, epithelial	European Collection of Authenticated Cell Cultures (ECACC, Porton Down, UK)	RPMI1640 + GlutaMAX containing 10% (*v*/*v*) FBS (GE Healthcare HyClone, Logan, UT, USA)
15	A2780/CDDP	ovary carcinoma, epithelial, resistant to cisplatin	European Collection of Authenticated Cell Cultures (ECACC, Porton Down, UK)	RPMI1640 + GlutaMAXcontaining 10% (*v*/*v*) FBS, additionally supplemented with 1 µM cisplatin
16	SKOV-3	ovary adenocarcinoma	American Type Culture Collection (Rockville, MD, USA)	McCoy’s 5A medium, supplemented with 10% (*v*/*v*) FBS (GE Healthcare HyClone, Logan, UT, USA)
17	MCF-7	mammary gland adenocarcinoma	European Collection of Authenticated Cell Cultures (ECACC, Porton Down, UK)	Eagle’s medium (HIIET, PAS, Wroclaw, Poland), supplemented with 10% (*v*/*v*) FBS (Sigma-Aldrich Chemie GmbH, Steinheim, Germany), 2 mM L-glutamine, MEM non-essential amino acid solution 1% (*v*/*v*) (Sigma-Aldrich Chemie GmbH, Steinheim, Germany), insulin 8 µg/mL (Sigma-Aldrich Chemie GmbH, Steinheim, Germany)
18	MDA-MB-231	mammary gland adenocarcinoma	American Type Culture Collection (ATCC, Manassas, VA, USA)	RPMI 1640 (HIIET, PAS, Wroclaw, Poland), supplemented with 10% (*v*/*v*) FBS (Sigma-Aldrich Chemie GmbH, Steinheim, Germany) and 2 mM L-glutamine
19	PC-3	prostate adenocarcinoma	European Collection of Authenticated Cell Cultures (ECACC, Porton Down, UK)	RPMI 1640, supplemented with 10% (*v*/*v*) FBS (GE Healthcare HyClone, Logan, UT, USA) and 2 mM L-glutamine

* All culture media were supplemented with antibiotics—100 µg/mL streptomycin (Polfa-Tarchomin, Warsaw, Poland) and 100 U/mL penicillin (Sigma-Aldrich Chemie GmbH, Steinheim, Germany).

**Table 3 sensors-22-06183-t003:** Set of compounds and their concentration ranges utilized for antiproliferative assay.

Compound	Concentration Ranges [µM]	Manufacturer
Paclitaxel	0.1–0.0001	Fresenius Kabi
Cisplatin	100–0.1	Accord
5-Fluorouracil	100–0.1	Accord
Etoposide	100–0.1	Sigma-Aldrich
Gemcitabine	0.1–0.001	Sigma-Aldrich
Doxorubicin	1–0.001	Sigma-Aldrich

**Table 4 sensors-22-06183-t004:** Characteristics of cancer cell growth (proliferation) on the LOC platform using a standard (cell line-optimized) culture medium.

Cell Line	Cell Line Type	Growth Rating After *		
		24 h	72 h	120 h
A549	lung carcinoma	++/++	++/++	−/++
A498	kidney carcinoma	++/++	++/+	+/++
5637	urinary bladder TCC	+/−	+/−	−/−
RT-112	urinary bladder TCC	++/++	+++/+++	+/+++
TCC-SUP	urinary bladder TCC	++/+	+++/++	+++/+++
UM-UC-3	urinary bladder TCC	++/+	++/++	+/+++
UM-UC-3/CDDP	urinary bladder TCC, resistant to cisplatin	++/−	+/−	−/−
UM-UC-3/GEM	urinary bladder TCC, resistant to gemcitabine	−/−	−/−	−/−
UM-UC-3/VBL	urinary bladder TCC, resistant to vinblastine	++/−	−/−	−/−
HCT116	colon carcinoma	++/+	++/++	+/+++
HT29	colon adenocarcinoma	++/++	+++/++	+/++
LoVo	colon adenocarcinoma	++/++	+/+++	+/+++
LoVo/DX	colon adenocarcinoma, resistant to doxorubicin	+/+	+/−	+/−
A2780	ovary carcinoma, epithelial	+/+	+/−	+/−
A2780/CDDP	ovary carcinoma, epithelial, resistant to cisplatin	−/+	−/−	−/−
SKOV-3	ovary adenocarcinoma	+++/+++	+++/+++	−/++
MCF-7	mammary gland adenocarcinoma	+/+	++/+	++/+++
MDA-MB-231	mammary gland adenocarcinoma	+/−	+/+	++/++
PC-3	prostate adenocarcinoma	+/+	++/+	++/++

x/y—cells seeded high/low density (5 × 10^4^ and 10^4^, respectively); * +++—high attachment rate, visible signs of proliferation and colonies formation; ++—low attachment rate, no signs of proliferation nor colonies formation; +—no attachment, but no signs of cell death; −—significant signs of cell death. Tests were performed in reference to control cultures representing +++/+++.

**Table 5 sensors-22-06183-t005:** Characteristics of cancer cell growth (proliferation) on the LOC platform utilizing CO_2_-independent medium.

Cell Line	Cell Line Type	Growth Rating After *		
		24 h	72 h	120 h
A549	lung carcinoma	++/++	++/++	−/++
A498	kidney carcinoma	+/+	++/+	+/++
RT-112	urinary bladder TCC	++/++	+++/+++	+/+++
TCC-SUP	urinary bladder TCC	+/+	+++/++	+++/+++
UM-UC-3	urinary bladder TCC	++/++	++/++	+/+++
HT29	colon adenocarcinoma	+/+	+++/++	+/++
LoVo	colon adenocarcinoma	−/+	−/++	−/++
SKOV-3	ovary adenocarcinoma	++/++	++/++	+/++
MCF-7	mammary gland adenocarcinoma	+/+	++/+	++/+++
MDA-MB-231	mammary gland adenocarcinoma	+/−	+/+	+/++

x/y—cells seeded high/low density (5 × 10^4^ and 10^4^, respectively); * +++—high attachment rate, visible signs of proliferation and colonies formation; ++—low attachment rate, no signs of proliferation nor colonies formation; +—no attachment, but no signs of cell death; −—significant signs of cell death. Tests were performed in reference to control cultures representing +++/+++.

**Table 6 sensors-22-06183-t006:** Potentially prevailing environmental conditions during the selected steps of the biological nanosatellite mission with a view to our experimentation.

Step	Phase	Environmental Conditions (Temperature and Relative Humidity)	Time
0	Payload processing facility	17–25 °C, 40–60%	~2–3 weeks
1	Launchpad activities	4–27 °C, ≤60%	~1–2 days
2	Flight	Payload will be exposed to an equivalent radiative heat flux emanating from about 93 °C surface with an emissivity of 0.9	~1 h
3	Free-orbiting satellite (space experimentation)	−120 ÷ +120 °C (outer space) [43]	~5 days
Overall mission time		~20–28 days	

## Data Availability

The authors declare that all data supporting the findings of this study are available within this article or from the corresponding author upon reasonable request.

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
