# Peer review of "Microfluidic-Assisted Human Cancer Cells Culturing Platform for Space Biology Applications"

_sensors, 2022, doi:10.3390/s22166183_

Round 1

Reviewer 1 Report

Review of the manuscript entitled:

Microfluidic-assisted human cancer cells culturing platform for space biology applications

This work describes the development of a novel lab-on-chip (LOC) platform that is intended to overcome the current limitations related to the long-term cultivation of human cancer cells in CubeSat biological missions. After detailing the characteristic of the proposed microfluidic solution, the authors performed laboratory tests to evaluate the advantages and limitations of the current design. In light of the promising results, further developments of the LOC platform are finally proposed.

The article is well structured and easy to follow. In spite of being at a early development stage, the proposed microfluidic solution seems to clearly overcome many of the issues that so far have limited the study of human cancer cells in microgravity conditions (space biology applications). As such, I believe its content could be of interest for the scientific community and more in particular to the readers of the Sensor journal. Therefore, I suggest its publication after a few minor revisions that are mostly concerning the introduction section. Details are provided below:

General:

·         The article could benefit from a detailed English revision, since several grammar errors are presented. As a representative example, I recommend checking the verb tenses used in the text: some sections are written in present perfect (e.g. lines 153-167), while others in past simple and past perfect.

·         When an acronym is introduced, it should be then use all along the text. For instance, LOC is introduced in the abstract section, and should replace “lab-on-chip” in the rest of the manuscript. In the current version, LOC and lab-on-chip are randomly used depending on the section.

·         The authors need to revise the numbering of the sections (3 sections are numbered as 2.3.2).

Abstract:

·         Please provide the LOC acronym (between parenthesis) on line 15, after lab-on-chip.

·         Please include a last sentence at the end of the abstract, briefly describing the content of section 3.4

Introduction:

·         Although providing key information to understand the proposed work and its context, I believe this section could benefit from a reorganization of its content. For instance, I believe the text between lines 81 and 85 should be provided in Materials and method rather than in the introduction. Furthermore, the authors should include an additional section describing previous works based on the laboratory or space application of LOC platform for the study of cancer cells (highlighting the novelty of this research compared to previous works).

Materials and methods:

·         Section 2.3.1: please specify that you are studying mammalian cancer cells (and more specifically humans) to better connect this section with the introduction.

Result:

·         Figure 3: please provide the definition of G1, S and G2 in the caption figure.

·         Lines 405-409: the results of the experiment shown in Figure 15 are not well described. Therefore, I suggest to add an additional paragraph explaining more in detail the reasons behind the observed results. Furthermore, it would be nice to describe the idea the authors have for improving the reliability of the LOC platform.

Author Response

The Authors thank the Reviewer for all the positive comments and interesting remarks. Responses to each of the statement have been addressed and are given below.

Reviewer’s remarks and Authors’ responses:

Review of the manuscript entitled: Microfluidic-assisted human cancer cells culturing platform for space biology application

This work describes the development of a novel lab-on-chip (LOC) platform that is intended to overcome the current limitations related to the long-term cultivation of human cancer cells in CubeSat biological missions. After detailing the characteristic of the proposed microfluidic solution, the authors performed laboratory tests to evaluate the advantages and limitations of the current design. In light of the promising results, further developments of the LOC platform are finally proposed.

The article is well structured and easy to follow. In spite of being at a early development stage, the proposed microfluidic solution seems to clearly overcome many of the issues that so far have limited the study of human cancer cells in microgravity conditions (space biology applications). As such, I believe its content could be of interest for the scientific community and more in particular to the readers of the Sensor journal. Therefore, I suggest its publication after a few minor revisions that are mostly concerning the introduction section. Details are provided below:

General:

  • The article could benefit from a detailed English revision, since several grammar errors are presented. As a representative example, I recommend checking the verb tenses used in the text: some sections are written in present perfect (e.g. lines 153-167), while others in past simple and past perfect.
  • The Authors thank for that comment. The manuscript underwent a detailed English revision. The tenses were corrected, i.e. in most cases Past Simple and Present Simple were used. Present Perfect was employed solely in the context of literature reports mentioning, e.g. “Investigation of the influence of microgravity onto biological and biomedical samples has been the subject of intensive scientific works for the last decades [1-3].”
  • When an acronym is introduced, it should be then use all along the text. For instance, LOC is introduced in the abstract section, and should replace “lab-on-chip” in the rest of the manuscript. In the current version, LOC and lab-on-chip are randomly used depending on the section.
  • Certainly, that is correct. However, to be consistent with the MDPI regulations, the Authors improved the manuscript and do not use acronyms in the Abstract section at all. At first appearance of the word “lab-on-chip” in the Introduction section, Authors introduce its abbreviation (LOC) and then use only this form.
  • The authors need to revise the numbering of the sections (3 sections are numbered as 2.3.2).
  • The Authors thank for the comment. The section numbering was corrected.

Abstract:

  • Please provide the LOC acronym (between parenthesis) on line 15, after lab-on-chip.
  • Please see the second Authors’ response. The acronym was provided in the line 100 (at first appearance in the text).
  • Please include a last sentence at the end of the abstract, briefly describing the content of section 3.4
  • Certainly, such sentence was included in the manuscript abstract: “At the end of the manuscript, the Authors provide the considerations regarding potential 3-Unit CubeSat biological mission launched with Virgin Orbit company. The lab-on-chip platform was modelled to fit 2-Unit autonomous laboratory payload.”

Introduction:

  • Although providing key information to understand the proposed work and its context, I believe this section could benefit from a reorganization of its content. For instance, I believe the text between lines 81 and 85 should be provided in Materials and method rather than in the introduction. Furthermore, the authors should include an additional section describing previous works based on the laboratory or space application of LOC platform for the study of cancer cells (highlighting the novelty of this research compared to previous works).
  • We thank the Reviewer for the comment. The aforementioned text was shifted to the Materials and Methods section. Moreover, the description of the previous works conducted by the group in the field of microfluidics cancer cell culturing platforms was introduced to the manuscript Introduction (line 90-96): “Preliminary research related to cancer cell cultures on-chip has been recently done by the group and published in the paper [33]. Herein, two cell lines were investigated, i.e. human keratinocytes (HaCaT) and skin melanoma (A375) by culturing in LOCs under stationary conditions for 72 hours. This work is an improvement over the mentioned one in the context of: number of cancer cell lines for which the technology was validated, culturing duration, harsh culturing conditions applied (ambient temperature), and usage of different LOC geometry and microflow perfusion system.”

Materials and methods:

  • Section 2.3.1: please specify that you are studying mammalian cancer cells (and more specifically humans) to better connect this section with the introduction.
  • That is right, this information was added at the beginning of the Section 2.3.1: “The human cancer cell lines used in the study are summarized in Table 2 along with the culture media used for cells culturing.” (Line 188-189).

Result:

  • Figure 3: please provide the definition of G1, S and G2 in the caption figure.
  • We thank for that note. The missing definitions of the Fig. 14 were provided. G1 stands for the cell growth stage, S for the DNA synthesis stage and G2 for pre-mitosis stage (Line 425-426).
  • Lines 405-409: the results of the experiment shown in Figure 15 are not well described. Therefore, I suggest to add an additional paragraph explaining more in detail the reasons behind the observed results. Furthermore, it would be nice to describe the idea the authors have for improving the reliability of the LOC platform.
  • The Authors thank for the comment. More detailed information regarding the biological state of the cells was added, as well as about the methods to improve the cells growth on-chip utilizing perfusion system (Line 435-448).

“In reference to the Fig. 15, cells morphology differed in comparison to previous tests: cells were rounded with membranes surrounded by some number of debris,
nevertheless, the cells growth was sustained with relatively high percentage of cells
attachment. Although the micro flow culture of the human cancer cells still needs to be improved, those two factors seems crucial for further studies in microgravity.

One of the modification of the microflow system to enhance the proper cells growth could be at first the application of more physiological, smaller flow rate value (<250 µL/min) – not introducing excessive shear forces which can influence cells morphology badly. Different micropump could be used for this purpose, e.g. RP-TX, Takasago Fluidic Systems, Japan, which operates at nL-µL range. Other changes could be related to LOC structure. For instance, microchannel depth could be bigger, ensuring greater space for colony growth. Further suggestion is to apply a matrix of small via holes, being coupled with microchannel to allow for more effective gas exchange within the culture area. This solution could prevent the cells from hypoxia and related unphysiological growth.”

Reviewer 2 Report

Authors proposed a novel prototype of the lab-on-chip platform that is able to cultivate human cancer cells in vitro in long term. The prototype were presented in details and tested with 19 types of human cancer cells. Various combination of cultivation condition were tested and results showed that most cell lines showed at least acceptable morphology. I recommend publish the paper with a few major revisions.

1.       A control group of cancer cells cultivated in regular lab device is missing. Comparing cell morphology and attachment rate between the proposed lab-on-chip platform and the regular cultivation system is the direct evidence to prove validity of the prototype.

2.       In table 4 and table 5, how is cells attachment rate evaluated? Is it purely based on observation and subjective judgement, or based on any quantification method?

3.       “The major assumption of the paper is to propose the dedicated lab-on-chip system that would allow for cultivation of the cells in ambient conditions, without perfusion system (imitating conditions prior to rocket launch) and next, ensuring “regular” culturing experimentation, i.e. utilizing automated medium delivery system and standard incubation temperature (when the nanosatellites modules are initialized on the orbit).”

Is it an assumption or a proposal? Make word clear to understand.

4.       All the figures have low resolution. Both images and figures are blurred. Authors need to improve figure quality.

5.       What is the power consumption of the lab-on-chip system? Is the system powered by outlet or batteries?

6.       What is the required power consumption prior and during the launch mission?

7.       Figure 2.b shows the prototype, however, it’s not well packaged. For the launch mission, what packaging standard is required?

Author Response

The Authors thank the Reviewer for all the positive comments and interesting remarks. Responses to each of the statement have been addressed and are given below.

Reviewer’s remarks and Authors’ responses:

Authors proposed a novel prototype of the lab-on-chip platform that is able to cultivate human cancer cells in vitro in long term. The prototype were presented in details and tested with 19 types of human cancer cells. Various combination of cultivation condition were tested and results showed that most cell lines showed at least acceptable morphology. I recommend publish the paper with a few major revisions.

  1. A control group of cancer cells cultivated in regular lab device is missing. Comparing cell morphology and attachment rate between the proposed lab-on-chip platform and the regular cultivation system is the direct evidence to prove validity of the prototype.

Ad. 1. We thank the Reviewer for that comment. All the research (especially search for the most appropriate cell line to be cultured on-chip) was complemented by the reference cultures established at Petri dishes or culture flasks in corresponding laboratory conditions. Cells growth on-chip in comparison with control samples was assessed by an experienced scientist from Laboratory of Experimental Anticancer Therapy, Hirszfeld Institute of Immunology and Experimental Therapy, Polish Academy of Sciences.

This information was added to all the necessary manuscript sections. Please see the example lines: 276-277, 315-317, as well tables (Table 4 and 5). In these tables (treating about cells growth rate), the missing statement was introduced, i.e. “Tests were performed in reference to control cultures representing +++/+++”

  1. In table 4 and table 5, how is cells attachment rate evaluated? Is it purely based on observation and subjective judgement, or based on any quantification method?

      Ad. 2. The evaluation of the cells growth/attachment was based on visual observation conducted by experienced scientists, with the cells cultured in standard laboratory devices used as a reference. No quantitative methods were applied in that case. Please, remember that quantitative analysis of cells growth at this stage was not our goal, but to choose the most robust cell lines for further studies.

  1. “The major assumption of the paper is to propose the dedicated lab-on-chip system that would allow for cultivation of the cells in ambient conditions, without perfusion system (imitating conditions prior to rocket launch) and next, ensuring “regular” culturing experimentation, i.e. utilizing automated medium delivery system and standard incubation temperature (when the nanosatellites modules are initialized on the orbit).”

Is it an assumption or a proposal? Make word clear to understand.

      Ad. 3. The Authors thank for that comment. Certainly, the word was changed to the suggested by the reviewer to be clear:

      “The major proposal of the paper is to present the dedicated LOC system that would allow for cultivation of the cells in ambient conditions, without perfusion system (imitating conditions prior to rocket launch) and next, ensuring “regular” culturing experimentation, i.e. utilizing automated medium delivery system and standard incubation temperature (when the nanosatellites modules are initialized on the orbit).” (Line 81-85).

  1. All the figures have low resolution. Both images and figures are blurred. Authors need to improve figure quality.

      Ad. 4. That is truth. All the figures were improved. The captions provided in the images’ area were also corrected.

  1. What is the power consumption of the lab-on-chip system? Is the system powered by outlet or batteries?

      Ad. 5. The power consumption of the lab-on-chip system is circa 2 W. 1 W is needed for the micropump operation, additional 1 W is used for the micropump controller. The system is powered by 12 V power supply, thus laboratory version does not use batteries. Different approach is going to be applied in the lab-payload construction (please see the Ad. 6).

      The information about power consumption of the lab-on-chip system was added to the manuscript (line 126-128).

  1. What is the required power consumption prior and during the launch mission?

Ad. 6. Prior to the biological mission assuming cell culturing of cancer cells in microfluidic laboratory payload, the power consumption can be equal to 0 W. This is the case when the temperature at the Launchpad (Phase 1) is ambient (Please, see the Table 6). If the temperature is lower, a power supply can be provided on the request by the integrator to maintain the ambient temperature within the payload. In this case the power consumption should not exceed 4 W.

During the mission, the maximum continuous power consumption of the payload is circa 8 W (estimation provided based on the laboratory tests). Typically for such space missions, the payload is powered by a special satellite 3S3P battery pack (nominal voltage of 3.6 V and capacity of 18.2 Wh for each cell), included in the Electrical Power System (EPS) – a module of the satellite base.

The information about the power consumption of the nanosatellite during the space mission was added to the manuscript (line 472-473).

  1. Figure 2.b shows the prototype, however, it’s not well packaged. For the launch mission, what packaging standard is required?

      Ad. 7. Yes, the Fig. 2b shows the laboratory microfluidic system that has not been packaged yet. To appropriately prepare the payload for the biological mission, a special support structure, so called scaffold, is required to arrange all the components rigidly. In the case of our payload concept (Fig. 16), the support structure was designed to be fabricated out of PEEK, in the milling process. PEEK was chosen due to, for instance, its high heat resistance (continuous service temperature: 260 ºC), as well as notable resistance to radiation [1]. PEEK, similarly as polyimide, is a lightweight, “space material”, which exhibits substantial stability at high vacuum conditions. Utilizing the PEEK scaffold, LOCs, media reservoirs, micropumps, cameras, etc. can be reliably assembled.

      The scaffold including all the components is called “insert”. Next, the insert is placed into the housing (in our case it is a polished aluminum), the surface of which is adapted for integration into the cubic shape of the satellite platform in the CubeSat standard.

      The information concerning the PEEK scaffold used for our laboratory payload concept, as well as aluminum housing standard was added to the manuscript text (line 456-457, 466-467) and indicated in the Fig. 16.

[1] S. Kalra, et al., Investigations on the suitability of PEEK material under space environment conditions and its application in a parabolic space antenna, Advances in Space Research, vol. 63, pp. 4039-4045, 2019.

Round 2

Reviewer 2 Report

Authors haved addressed all questions properly.

The cells attachment rate is evaluated by observation,
which can be improved further, as observation itself is not reliable and different scientists may give inconsistent results.

Overall, the manuscript is in a high quality. I recommned an acceptance.